# Measuring job satisfaction of midwives: A scoping review

Sonja Wangler[1,2]*, Joana Streffing[1], Anke Simon[2], Gabriele Meyer[1], Gertrud M. Ayerle[1]

**1** Institute of Health and Nursing Science, Medical Faculty, Martin Luther University Halle-Wittenberg, Halle (Saale), Germany, **2** School of Business and Health, Baden-Wuerttemberg Cooperative State University (DHBW), Stuttgart, Germany

* Sonja.wangler@dhbw-stuttgart.de

## Abstract

### Background

Given the global shortage of midwives, it is of utmost interest to improve midwives' job satisfaction and working environments. Precise measurement tools are needed to identify both predictors of job satisfaction and intervention strategies which could increase it. The aim of this study is to collate, describe and analyse instruments used in research to assess the job satisfaction of midwives working in hospitals, to identify valid and reliable tools and to make recommendations for the further development of specific instruments for midwifery practice and future midwifery research.

### Methods

We conducted systematic literature searches of the following databases: CINAHL, MEDLINE, PsycINFO, Web of Science Core Collection, Cochrane Database. Studies which assessed the job satisfaction of midwives working in a hospital setting were eligible for inclusion.

### Findings

Out of 637 records 36 empirical research articles were analysed, 27 of them cross-sectional studies. The studies had been conducted in 23 different countries, with sample sizes ranging between nine and 5.446 participants. Over 30 different instruments were used to measure midwives' job satisfaction, with considerable differences in terms of domains evaluated and number of items. Twelve domains relevant for job satisfaction of midwives working in hospitals were identified from the empirical studies. Four instruments met the defined reliability and validity criteria.

### Conclusion

Autonomy, the significance of the job, the challenges of balancing work and private life, and the high emotional and physical demands of midwifery are job characteristics which are underrepresented in instruments measuring job satisfaction. The influence of the physical

**Data Availability Statement:** All relevant data are within the paper and its Supporting Information files.

**Funding:** The authors received no specific funding for this work.

**Competing interests:** The authors have declared that no competing interests exist.

working environment has also not yet been researched. There is a need to develop or adapt instruments to the working environment of midwives.

## Introduction

Maternity care in hospitals is highly dependent on the midwifery workforce in many countries. However, the worsening global shortage of midwives and resultant vacant positions in labour wards puts the quality of care for mothers, babies and their families at risk [1, 2]. Job satisfaction of midwives and other health care personnel is an important factor influencing not only personal wellbeing, commitment and workforce retention but also work performance and outcomes. Improving midwives' job satisfaction is one intervention to keep midwives in the profession and counteract midwife shortages [3–5].

Job satisfaction is described as a comprehensive concept made up of various components, with overall satisfaction being the cumulative result of these components [6]. The relationship between components of job satisfaction and overall job satisfaction is explained in several job satisfaction theories. The Job Characteristic Model by Hackman and Oldham [7] and Herzberg's two factor theory [8], both belonging to the motivational approach, describe the important role of intrinsic aspects (individual needs for growth, development and the meaningfulness of the work) in job satisfaction. Humphrey et al. explore this motivational approach in more depth, emphasising the importance of social characteristics (support, interaction) and work context (work environment, ergonomics, noise) [9].

Research has identified several approaches to improving the job satisfaction of midwives working in hospitals. Important motivators which positively influence midwives' job satisfaction are support within the team, good relationships with colleagues [1, 10–12], appreciation and support from superiors [2, 13–15], autonomy, meaningfulness of the work, interaction with women, and being able to support normal birth [3, 5, 16–19]. Factors which reduce job satisfaction are heavy workload, lack of staff and resources, conflicts in work-life balance and low salary [4, 14, 16, 20–22].

Valid and psychometrically sound measuring instruments are needed to evaluate intervention strategies designed to improve job satisfaction and the working environment.

Numerous instruments exist, in particular questionnaires, developed through research on job satisfaction in organisational psychology—some for jobs in general, others for specific jobs [6, 23]. Most questionnaires assess job satisfaction multi-dimensionally, looking at several components, others measure global job satisfaction [24]. There is no common standard as to which work aspects or dimensions should be considered or which questionnaire should be used [24]. Our aim therefore is to collate, describe and analyse instruments used in research to assess the job satisfaction of midwives working in hospitals, to identify valid and reliable tools, and to make recommendations both for the further development of instruments specific to midwifery practice and for future midwifery research.

## Methods

We conducted a scoping review in order to explore the extent of the literature in the field of midwifery job satisfaction and to examine how research in this field is conducted. Scoping reviews aim to identify and map available evidence on an area of research in a transparent way [25–27]. They bring together the evidence from heterogeneous sources and study approaches and can therefore detect research gaps in the existing literature [26, 28]. The Joanna Briggs

Institute's Methodological Guidance [29] was followed, based on work by Arksey and O'Malley [26]. The PRISMA Extension for Scoping reviews (PRISMA-ScR) was used to structure this article [30].

### Eligibility criteria

We included journal publications of studies in English or German which quantitatively or qualitatively assessed the job satisfaction of midwives working in a hospital setting. In order to obtain recent and transferable results, we limited the time period to studies that had been published from 2010 onwards. The study sample had to include at least 50% midwives. Studies focusing mainly on such concepts as burnout, work engagement or stress, rather than job satisfaction, were excluded. Also excluded were studies which focused on the situation of midwifery trainees/students, as they often have a different perspective. Instruments for which no validation study could be found were excluded.

### Sources of information and search

We conducted a systematic literature search including database searches (CINAHL, MEDLINE via Pubmed, PsycINFO via Ovid, Web of science core collection, and Cochrane Database), free web searching and backward and forward citation. The 'Population, Concept, Context' (PCC) Criteria (according to the Prisma-ScR [30]) were used to develop the search string. The search terms used were midwife, midwives, midwifery AND hospital, obstetric, ward, unit, department, obstetrical. They were combined using AND with synonyms for the concept of job satisfaction: job satisfaction, quality of work life, work satisfaction, employee satisfaction, and with synonyms for the data assessment: questionnaire, instrument, scale, measurement, assessment, appraisal, evaluation, interview and focus group (see S1 File).

### Selection of sources and data charting process

One reviewer (SW) designed and conducted the search strategy supported by the second reviewer (GMA). Two reviewers (SW, JS) independently screened titles, abstracts, and full-text articles for inclusion. A data extraction sheet for the compilation of content was created by SW following the JBI manual [29]. The characteristics extracted included: country, study design and objectives, context, population and sample size, with a focus on tools measuring midwives' job satisfaction. The instruments were assessed based on the following key information: type of instrument, theoretical background, dimensions and items, response scales, reliability and validity. If the items or information about the questionnaire were not listed in the article, their development and validation studies were procured for further data extraction.

### Assessment of reliability and validity

The reliability of the instrument was assessed by means of internal consistency (Cronbach's alpha). An instrument with an internal consistency coefficient of 0.80 or higher was considered good [31].

To find out whether the instruments were applicable to midwives in hospitals, we checked whether the entire construct of job satisfaction was represented (content validity). The domains identified by Van Saane et al. [24] in a systematic review were followed and compared with the factors found in the systematic literature search. Content validity was rated satisfactory if the instrument covered at least seven of twelve domains.

## Results

### Search and study selection

The search yielded a total of 626 records. After removing duplicates, all articles (n = 499) were transferred to the Covidence tool for systematic reviews [32] and screened using the above-mentioned inclusion and exclusion criteria. Ultimately, 96 full text articles were reviewed, and subsequently 60 further papers excluded, leaving 36 for further review and data extraction (Fig 1).

### Study characteristics

Characteristics, such as origin of the studies, publication year, and sample sizes in total are summarised in Table 1. Table 2 presents the included studies, their study design samples and objectives. Of the 36 publications, 27 research papers are descriptive and cross-sectional studies, two are longitudinal observational studies, two mixed-method studies and five are qualitative studies. In the quantitative studies, the average number of participants was 576 and ranged

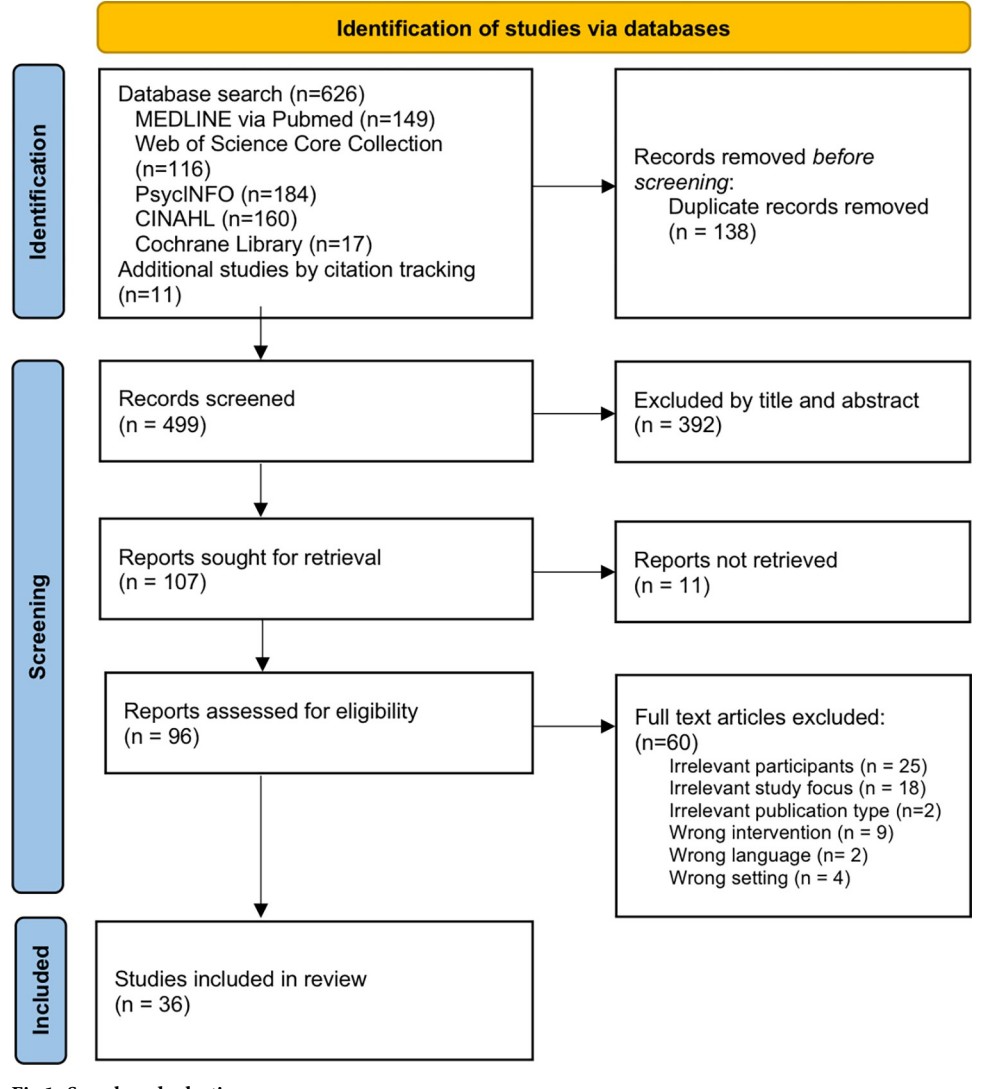

**Fig 1. Search and selection process.**

**Table 1. Study characteristics (*n* = 36).**

| | |
|---|---|
| Country | European Countries (n = 14) |
| | Australia / New Zealand (n = 8) |
| | North America (n = 2) |
| | African Countries (n = 5) |
| | Asian Countries (n = 4) |
| | Multinational with ≥ two countries (n = 3) |
| Year of publication | 2010–2015 (n = 13) |
| | 2016–2022 (n = 23) |
| Study design | Qualitative design (n = 5) |
| | Quantitative design (n = 29) |
| | Mixed-methods design (n = 2) |
| Participants (in total) | n = 17.957 |

from 43 to 5.446. In the qualitative studies, the average number of participants was 16 and ranged from nine to 26.

## Characteristics of the construct *midwives' job satisfaction*

The authors Van Saane et al. designate job characteristics which form the basis for the construct job satisfaction in general [24]. They categorise those job characteristics in eleven domains: *work content, autonomy, growth/development, financial rewards, promotion, supervision, communication, co-workers, meaningfulness, workload,* and *work demands.* These domains were confirmed by our literature search. The domains *co-workers, meaningfulness, work content, autonomy* and *workload* proved to be very important for midwives [4, 10, 12, 13, 20, 22, 35, 38, 41, 43, 44, 52]. Due to overlap of the domains *communication* and *co-workers,* and of *development* and *promotion,* we combined these domains into *working relationships* and *growth/development.* However, three other factors which influence midwives' job satisfaction emerged from the papers reviewed which were not represented in the domains identified so far. One factor is the *physical working environment* of midwives in the hospital (in terms of materials and equipment, as well as the influence of atmosphere and room design) [17, 18, 34, 36], a second is staff *health* [10, 42, 45]. Furthermore, the aspect of *work-life balance* appeared to be important for midwives and influenced their job satisfaction [11, 16, 39, 40]. We added these factors and assessed the content validity of the instruments using 12 domains (Fig 2).

## Instruments

In total, 35 different instruments were identified in the included studies. Researchers often combined up to six different assessment instruments by supplementing generic questionnaires with questionnaires examining related constructs of job satisfaction [1, 2, 10, 14, 33, 35, 38, 40, 46, 47, 50]. All studies used self-administered questionnaires as preferred research tools, with items rated on Likert Scales (4- to 7-point Likert Scales). The number of items varied considerably, ranging from 20 to 77, especially if different instruments were used. The tools can be divided into three categories: 1) Global job satisfaction instruments, 2) Multi-dimensional (faceted) job satisfaction instruments, and 3) Instruments measuring (single) components of job satisfaction.

**1) Global instruments.** Global instruments consider job satisfaction to be a global construct and ask directly about general feelings about the job to assess employees' overall job satisfaction, either in a single-or multiple-item version [53]. Only two of the instruments used in

**Table 2. Included studies (*n* = 36).**

| Author (Year) Location | Journal | Study design Sample | Aim of study |
|---|---|---|---|
| Adolphson (2016) [18] Mozambique | Midwifery | Qualitative study Midwives in different work settings, N = 9 | To explore midwives' perspectives on their working conditions and their professional role in a low-resource setting |
| Alnuaimi (2020) [1] Jordan | International Nursing Review | Cross-sectional study Midwives in hospitals & health centres; N = 413 | To assess the levels of Jordanian midwives' job satisfaction, intention to stay and work environment |
| Arefi (2021) [33] Iran | Pakistan Journal of Medical and Health Sciences | Descriptive study Midwives in two hospitals; N = 143 | To examine the relationship between job satisfaction, mental workload, and job control in hospital midwives |
| Bekru (2017) [11] Ethiopia | PLOS ONE | Cross-sectional study Midwives in hospitals & health centres; N = 221 | To assess job satisfaction and factors associated with same |
| Bourgeault (2012) [34] Canada | Midwifery | Qualitative study Community midwives (home & hospital) N = 26 | To explore the implications of midwives' place of work on their experiences as workers |
| Carolan-Olah (2015) [17] Australia | Midwifery | Qualitative study Hospital midwives N = 22 | To explore midwives' experiences of factors which facilitate or impede midwifery practice |
| Casey (2010) [35] Ireland | Journal of Nursing Management | Cross-sectional study Nurses & midwives N = 244 | To test an expanded model of empowerment and the impact on job satisfaction |
| Cronie (2019) [20] Netherlands | BMC Health Services Research | Cross-sectional study Hospital & primary care midwives; N = 508 | To measure job satisfaction of midwives and compare satisfaction levels between hospital and primary-care midwives |
| Davis (2016) [36] Australia and UK | Women and Birth | Qualitative study Midwives (home & hospital setting) in Australia & UK N = 12 | To examine the impact of the workplace on midwives |
| Direkvand-Moghadam (2022) [37] Iran | PLOS ONE | Mixed-method study Midwives in hospitals & health centres; N = 121 | To design a valid and reliable instrument to assess Iranian midwives' job satisfaction |
| Freeney (2013) [38] Ireland | Journal of Health Organization and Management | Cross-sectional study Midwives & nurses; N = 158 | To investigate work engagement and its influence on quality of care and general health of midwives |
| Geuens (2015) [10] Belgium | Nursing Management | Cross-sectional study Hospital midwives; N = 192 | To explore burnout, job satisfaction and intention to leave |
| Grylka-Baeschlin (2022) [39] Switzerland | BMC Health Services Research | Longitudinal observational study Hospital midwives; N = 43 | To assess job satisfaction before and after implementing a continuity of care model |
| Hildingsson (2015) [2] Sweden | Sexual & Reproductive HealthCare | Cross-sectional study Hospital midwives; N = 475 | To explore the practice environment of midwives and factors associated with the perception of an unfavourable work environment |
| Jarosova (2016) [40] European and Asian countries | Journal of Nursing Management | Cross-sectional study Hospital midwives; N = 1.190 | To investigate the relationship between turnover intentions and job satisfaction and the differences between countries |
| Jasiński (2021) [21] Poland | Medycyna Pracy | Cross-lagged survey Midwives in public health service; N = 225 | To evaluate correlations between workload, job satisfaction and stress before and during the COVID-19 pandemic. |
| Kalicińska (2012) [14] Poland | International Journal of Nursing Practice | Cross-sectional study Midwives & hospice nurses; N = 117 | To investigate the relationship between workplace support and burnout for midwives and hospice nurses |
| Khavayet (2018) [12] Iran | Journal of Midwifery & Reproductive Health | Cross-sectional study Hospital midwives; N = 100 | To evaluate the job satisfaction of midwives working in hospitals |
| Lumadi (2019) [19] South Africa | Curationis | Qualitative study Midwives in maternity wards; N = 11 | To explore the perceptions of midwives on the shortage and retention of staff at a public institution |
| Matthews (2021) [5] Australia | Women and Birth | Cross-sectional study Midwives in a tertiary hospital; N = 302 | To explore factors affecting Australian midwives' job satisfaction |

(*Continued*)

**Table 2.** (Continued)

| Author (Year) Location | Journal | Study design Sample | Aim of study |
|---|---|---|---|
| Mharapara (2021) [41] New Zealand | Women and Birth | Cross-sectional study Lead Maternity Carer midwives, employed midwives; N = 705 | To explore the effect of job characteristics on the job satisfaction of midwives practising in different work settings |
| Muluneh (2021) [22] Ethiopia | Women and Birth | Cross-sectional study Midwives; N = 107 | To analyse midwives´ job satisfaction and intention to leave their current position in developing regions of Ethiopia |
| Okuyucu (2019) [42] UK | Midwifery | Cross-sectional study Midwives 66% maternity unit; N = 635 | To investigate the musculoskeletal disorders of midwives and to explore individual, work-related and psychosocial risk factors |
| Pallant (2016) [43] New Zealand | Women and Birth | Cross-sectional study Hospital midwives; N = 600 | To explore the association between scores on the PES subscales and midwives' intention to leave the profession |
| Papoutsis (2014) [13] Greece | British Journal of Midwifery | Cross-sectional study Midwives in public & private hospitals; N = 145 | To examine the job satisfaction of hospital-practising registered midwives and determine the main predictors of job satisfaction |
| Perdok (2017) [44] Netherlands | Midwifery | Cross-sectional study Midwives (primary care & clinical), obstetricians, obstetric nurses; N = 799 | To assess how maternity care professionals perceive their job autonomy |
| Perry (2017) [45] Australia | Journal of Advanced Nursing | Cross-sectional study Nurses & midwives in different settings; N = 5.446 | To examine the quality of life of nurses and midwives and identify predictive factors of quality of life |
| Peter (2021) [16] Switzerland | BMC Health Services Research | Cross-sectional study Hospital midwives N = 98 | To investigate work-related stress and intentions to leave |
| Rodwell (2013) [46] Australia | Journal of Advanced Nursing | Cross-sectional study Hospital nurses & midwives; N = 273 | To investigate the relationship between job control, social support and organisational justice and the impact on job satisfaction |
| Rouleau (2012) [47] Senegal | Human resources for health | Longitudinal study Hospital midwives; N = 226 | To explore midwives' job satisfaction and its effects on burnout, intention to quit and professional mobility |
| Skinner (2012) [48] Australia | Australian Journal of Advanced Nursing | Cross-sectional study Nurses & midwives; N = 550 | To assess factors contributing to nurses' and midwives' job satisfaction |
| Stahl (2016) [15] Germany | Journal of Obstetric, Gynecologic & Neonatal Nursing | Cross-sectional study Hospital midwives; N = 1.692 | To describe the adaptation and psychometric testing of the Picker Employee Questionnaire |
| Sullivan (2011) [49] Australia | Midwifery | Cross-sectional study Hospital midwives; N = 209 | To determine factors contributing to the retention of midwives |
| Talasaz (2017) [50] Iran | Health Scope | Cross-sectional study Midwives of Mashad University; N = 107 | To determine the predictive power of job satisfaction and occupational stress in organisational commitment among midwives |
| Thumm (2020) [51] United States | Journal of Midwifery & Women's Health | Cross-sectional study Midwives in hospitals & medical centres; N = 2.333 | To test the validity and reliability of the newly designed Midwifery Practice Climate Scale |
| Vivilaki (2019) [52] Greece | Archives of Hellenic Medicine | Cross-sectional study Hospital midwives; N = 100 | To assess the working conditions of midwives and test the Greek translation and confirm its reliability and structural validity |

the studies reviewed were global instruments: The 'Satisfaction with Work Scale' (SWWS, developed by Diener et al. [54]) and the 'Overall Job Satisfaction Scale' (OJS, designed by Brayfield & Rothe [55]). They each used five items to assess global job satisfaction. Both are generic instruments with a Cronbach's alpha of 0.85 for SWWS [21] and 0.93 for OJS [41, 46], indicating good reliability for use with midwives. Both instruments were used in combination with single component instruments.

**2) Multi-dimensional (faceted) instruments.** Multi-dimensional or faceted instruments aggregate multiple items to different facets of job satisfaction. Faceted instruments represent the multi-dimensionality of the construct *job satisfaction* well [53]. Each facet may be presented with a single or multiple items. Multi-dimensional instruments allow statements about the influence of single items/facets of job satisfaction or their correlation with it. 13 of the

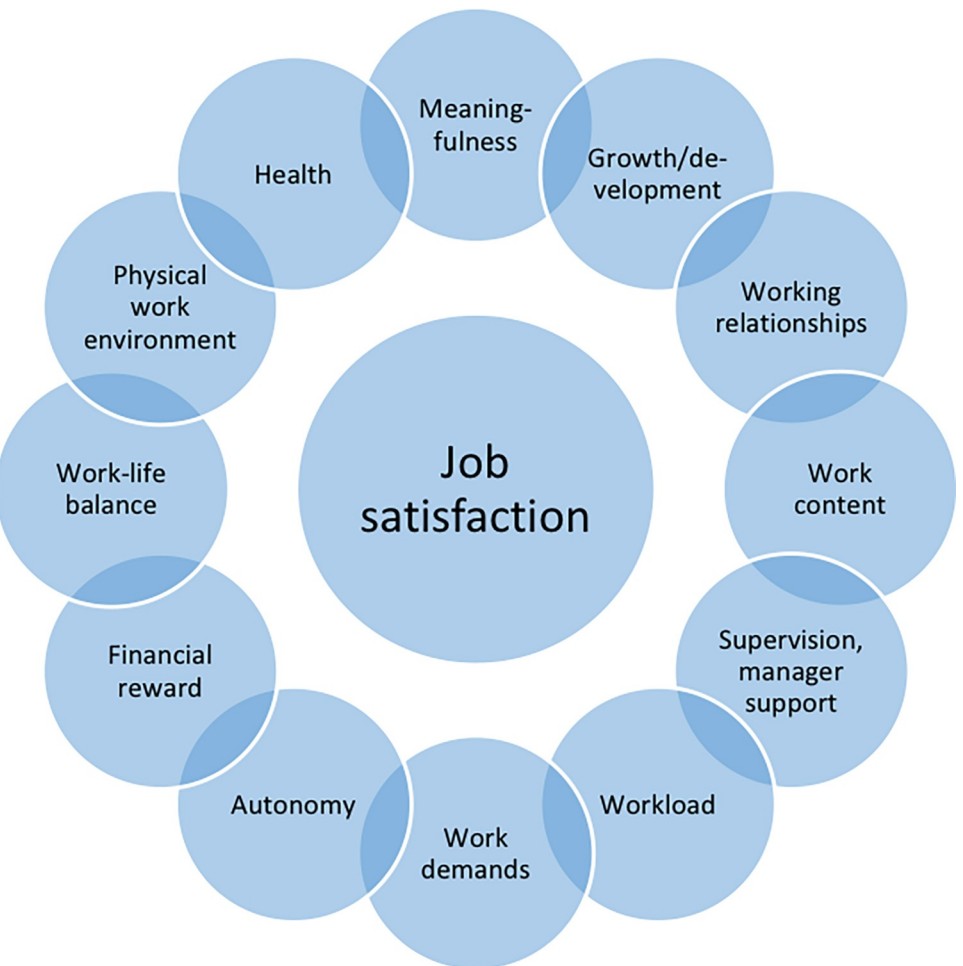

**Fig 2. Domains of midwives' job satisfaction.**

instruments presented were facet instruments, but they were heterogeneous in terms of the facets they depicted and the number of items. Table 3 gives an overview of these multi-dimensional instruments, their frequency of application, the theoretical foundation, reliability in the specific application of midwifery, the target group and the number of items and subscales. Those instruments with good reliability are printed in bold in the table. Table 4 depicts their content validity, instruments with good reliability and content validity in bold.

One tool was developed specifically for midwives: the Midwifery Process Questionnaire [68], focusing on midwives' view of their professional role. However, neither of the studies which used this questionnaire reported Cronbach's alpha for reliability [5, 39]. The following four instruments met the criteria for reliability and content validity and are therefore described in more detail. Two are generic instruments and two are instruments developed for the nursing profession.

*Generic instruments.* Minnesota Satisfaction Questionnaire-Short Form (MSQ-SF) and Job Satisfaction Scale (JSS).

Talasaz et al. [50] used the MSQ-SF, developed by Weiss et al. 1967 [61]. It measures job satisfaction on 20 facets, each with only one item, using a 4-point Likert response scale (1 = 'very dissatisfied' to 4 = 'very satisfied'). The MSQ-SF is a generic instrument that has been used for over 30 years in a wide range of jobs and is available in many languages [69]. The MSQ-SF

**Table 3. Multi-dimensional instruments.**

| Multi-dimensional instruments | Frequency of application [authors] | Theoretical foundation | Reliability (Cronbach's Alpha) | Developed for | Items | Sub-scales |
|---|---|---|---|---|---|---|
| **MMSS: McCloskey/Mueller Satisfaction Scale [56]** | **3 [1, 11, 40]** | | **0.92** | **nurses** | **31** | **8** |
| **LQWLQ–N: Leiden Quality of Work Life Questionnaire for Nurses [57]** | **3 [20, 39, 44]** | **Job Demand-Control-Support model [58]** | **0.81** | **nurses, adapted for maternity-care professionals** | **77** | **10** |
| **JSS: Warr´s Job Satisfaction Scale [59]** | **1 [35]** | | **0.88** | **generic use** | **17** | |
| GJSS: Generic Job Satisfaction Scale [60] | 1 [10] | | 0.71 | generic use | 10 | |
| **MSQ-SF: Minnesota Satisfaction Questionnaire-Short Form [61]** | **1 [50]** | | **0.85** | **generic use** | **20** | **2** |
| Job Satisfaction Questionnaire [62] | 1 [13] | Herzberg's two factor theory [8] | 0.50–0.81 | nurses, adapted for midwifery practice | 26 | 7 |
| Picker Employee Questionnaire [15] | 1 [15] | | 0.50–0.90 | hospital staff, adapted for midwives | 75 | 14 |
| CWEQ-II: Conditions of Work Effectiveness Questionnaire-II [63] | 1 [35] | Kanter's Theory on Structural Empowerment [64] | 0.68–0.88 | nurses | 19 | 6 |
| PES-(NWI): Practice Environment Scale of the Nursing Work [65] | 3 [1, 2, 43] | | 0.76–0.95 | nurses, adapted for midwives | 20–30 | 4–5 |
| MPQ: Midwifery Process Questionnaire [66] | 2 [5, 39] | | - | midwives | 20 | 4 |
| COPSOQ: Copenhagen Psychosocial Questionnaire [67] | 2 [16, 39] | | - | generic use | 19 | 6 |
| Job Satisfaction Instrument [47] | 1 [47] | | 0.7 | health professionals | 29 | 9 |
| Iranian Midwives Job Satisfaction Instrument (MJSI) [37] | 1 [37] | | 0.71 | midwives | 25 | 5 |

classifies satisfaction as related to either extrinsic or intrinsic aspects of the job. The items are summed up to identify overall satisfaction. It covers nine of the twelve domains outlined above, the missing items being *work-life balance*, *physical work environment* and *health*.

Casey et al. [35] measured job satisfaction using the Warr, Cook, Wall Job Satisfaction Scale (JSS, 1979). It is a generic and widely used instrument with 17 items, each with a response range from 1 ('I'm extremely dissatisfied') to 7 ('I'm extremely satisfied'). Warr et al. regard job satisfaction as employees' satisfaction with intrinsic and extrinsic factors of the job [59]. Nine categories are represented, but the *physical environment* is represented with only one item. Items about *meaningfulness* and *health* are missing.

*Instruments for the nursing profession*. Leiden Quality of Work Life Questionnaire for Nurses (LQWLQ–N) and McCloskey/Mueller Satisfaction Scale (MMSS).

Cronie et al., Perdok et al. and Grylka-Baeschlin et al. used the LQWLQ–N version [57] to assess job satisfaction [20, 39, 44]. This questionnaire is a specific version for nurses based on the generic Leiden Quality of Work Questionnaire [70], which measures the key concepts of the Job Demand-Control-Support model [58]. Cronie et al. reformulated the questions for maternity care professionals [20]. Job conditions were measured with 77 items in 10 subscales on a 4-point Likert Scale ranging from 1 ('totally disagree') to 4 ('totally agree'). One of the subscales with six items focuses directly on job satisfaction calculated as a mean of these six items. The other subscales represent the domains *personnel and organisation*, *work demands and tasks*, *autonomy*, *social support at work*, *working relationships*, *workplace agreements and referrals*, *potential for development*, *financial reward*, *influence of work on private life*. Only the categories *meaningfulness* and *health* are not represented.

**Table 4. Domains of multi-dimensional assessment instruments.**

| Domains | MMSS | LQWLQ | JSS | GJSS | MSQ-SF | JSQ | Picker | CWEQ II | PES-NWI | MPQ | COSPOQ | JSI | MJSI |
|---|---|---|---|---|---|---|---|---|---|---|---|---|---|
| **Work content** | | + | + | | + | + | | | + | | + | + | |
| **Meaningfulness** | | | | | + | | | | | | + | | + |
| **Growth/potential for development/ promotion** | + | + | + | + | + | + | + | + | + | + | + | + | + |
| **Working relationships (Co-workers/ communication)** | + | + | + | + | + | + | + | + | + | + | + | + | + |
| **Supervision, manager support, policy** | + | + | + | + | + | + | + | + | + | + | + | + | + |
| **Workload** | + | + | + | + | + | + | + | + | + | | + | + | + |
| **Work demands** | + | + | + | + | + | + | + | + | + | + | + | + | + |
| **Autonomy, responsibility** | + | + | + | | + | + | | | + | + | + | + | + |
| **Financial reward** | + | + | + | + | + | + | | | | | | + | + |
| **Work-life balance** | + | + | | | | | | | | | + | | + |
| **Physical work environment** | | + | + (one item) | | | | + | | | | + (one item) | + | |
| **Health** | | | + | | | | | | | | + | | + |
| **Total score** | 8 | 10 | 9 | 7 | 9 | 8 | 6 | 5 | 7 | 5 | 11 | 9 | 10 |

MMSS: Mc Closkey/Mueller Satisfaction Scale, LQWLQ: Leiden Quality of Work Life Questionnaire, JSS: Warr´s Job Satisfaction Scale, GJSS: Generic Job Satisfaction Scale, MSQ-SF: Minnesota Job Satisfaction Questionnaire-Short Form, JSQ: Job Satisfaction Questionnaire (Labiris) PES-NWI: Practice Environment Scale-Nursing Work, H-JSQ: Herzberg´s Job Satisfaction Questionnaire, CWEQ II: Conditions of Work Effectiveness Questionnaire II, MPQ: Midwives Process Questionnaire, COSPOQ: Copenhagen Psychosocial Questionnaire, JSI (Rouleau): Job Satisfaction Instrument, MJSI: Iranian Midwives Job Satisfaction Instrument

Bekru et al. [11], Alnuaimi et al. [1], and Jarosova et al. [40] used the MMSS, developed in 1990, for measuring job satisfaction among nurses. It contains 31 items in eight subscales and responses are given on a five-point Likert scale ranging from 1 ('very dissatisfied') to 5 ('very satisfied'). It is one of the most widely used scales in nursing research, in a variety of clinical and geographical settings [40]. The subscales are *satisfaction with extrinsic rewards*, *scheduling*, *family-work balance*, *co-workers*, *interactions*, *professional opportunities*, *praise and recognition*, *control and responsibility*. The domains *meaningfulness*, *work content*, *physical work environment* and *health* are missing.

**3) Component instruments.** Component instruments are defined as tools that measure (single) components of the construct job satisfaction or related concepts. 17 studies used component scales. Researchers combined different instruments or selected particular items from questionnaires and added either a multi-dimensional instrument or an item on overall job satisfaction. More than 20 different component scales (see Table 5) were used, measuring for example *social support at work*, *organisational support*, *work engagement*, *work climate*, *organisational commitment*, *psychological empowerment*, *work stress*, *social provision* and *health*. The instruments had up to four subscales and between four and 37 items. The reliability (Cronbach's alpha) was between 0.75 and 0.93. One of these instruments (the Midwifery Practice Climate Scale) was developed to measure midwives' perceptions of the supportiveness of their work environments. The other scales were generic or developed for hospital staff.

## Discussion

A large number of studies have been published on the job satisfaction of midwives working in hospitals in different countries since 2010. A great variety of instruments was identified with various dimensions and combinations of items and instruments. In particular, a large number of questionnaires measuring related constructs, such as *stress at work*, *organisational*

**Table 5. Instruments measuring components of job satisfaction.**

| Instrument [study] | Reliability (Cronbach's Alpha) | Items |
|---|---|---|
| Psychological Empowerment Scale [35] | >0.82 | 12 |
| Utrecht Work Engagement Scale (UWES) [38] | 0.71–0.90 | 9 |
| Organizational Support Scale [38] | 0.75 | 4 |
| Social Provisions Scale [38] | 0.70–0.83 | 12 |
| Perceived Organisational Support [38] | 0.93 | 8 |
| General Health Questionnaire (GHQ) [38, 46] | 0.82–0.91 | 12–21 |
| Nordic Musculoskeletal Questionnaire [42] | - | - |
| Organizational Commitment Scale [46, 50] | 0.83–0.84 | 24 |
| Social Support at Work Scale [14] | 0.92–0.93 | 16 |
| Perceived Stress Scale [21] | 0.77 | 10 |
| Perception of Empowerment (PEMS-R) [41] | 0.75–0.81 | 6 |
| Quantitative Workload Inventory [21] | 0.87 | 5 |
| Work Ability Index [39] | - | 7 |
| Karasek's Job control Scale [46] | 0.89 | 9 |
| Quantitative Workload Scale [46] | 0.73 | 11 |
| Positive and Negative Affectivity Scale (PANAS) [46] | 0.89 | 10 |
| Short Form 12 Health Survey (SF-12) [45] | 0.85–0.86 | 12 |
| McCains Intent to Stay Scale [1] | 0.91 | 5 |
| Culture/Climate Assessment Scale (CCAS) [52] | 0.87 | 37 |
| Effort Reward Imbalance Questionnaire-short form (ERI) [42] | | 16 |
| Midwifery Practice Climate Scale—revised [51] | 0.84–0.89 | 10 |

*commitment* or *work engagement* was found. Almost all research teams used a different instrument or combination of instruments, some slightly adapted to midwifery practice. Three of the instruments were used in three different studies: the McCloskey/Mueller Satisfaction Scale (MMSS), the Leiden Quality of Work Life Questionnaire for Nurses (LQWLQ-N) and the Practice Environment Scale of the Nursing Work (PES-NWI). The Midwifery Process Questionnaire (MPQ) and the Copenhagen Psychosocial Questionnaire (COPOQ) were each used twice.

The variety of instruments used suggests that none satisfactorily covers all domains of midwives' job satisfaction in the hospital. This is also confirmed by the results of our literature search. While the dimensions *work demands*, *workload*, *working relationships*, *financial rewards*, *development* and *supervision* are included in almost all questionnaires, other dimensions are underrepresented. Although the research findings show the importance of autonomy and the significance of the job for the satisfaction of midwives, items reflecting these (*meaningfulness* and *autonomy)* are missing in several questionnaires. The significance of these intrinsic aspects is well described in theories of job satisfaction [8, 9].

Another important dimension is the balance between work and private life. The combination of shift work in the hospital setting and frequent overtime due to staff shortages could lead to a work-life imbalance resulting in reduced job satisfaction [1, 16]. This dimension was only examined in four questionnaires. As personal wellbeing affects job satisfaction, and vice versa, it is important this is reflected in instruments measuring job satisfaction. Aspects such as mental health and physical disorders still play a minor role in questionnaires but seem to be a significant factor influencing job satisfaction in midwifery practice. On the one hand, midwifery work can be physically challenging, resulting in musculoskeletal disorders which subsequently lead to reduced job satisfaction [42, 45]. On the other hand, physical and mental

overload and dissatisfaction in the job can lead to health impairments and even burnout [10, 45]. Moreover, items relating to the physical working environment also play a minor role in questionnaires used in the studies reviewed. Apart from items about provision of equipment and facilities [12], no items were identified which assessed the influence of the birthing room environment on midwives' job satisfaction. Qualitative research data suggest that the design of the labour room influences the work of midwives, as different designs create different atmospheres which affect midwives' wellbeing [17, 34, 36]. While the influence of the architecture and design of the birthing room on women giving birth has already been qualitatively researched [71–73], no quantitative studies were identified which assessed the environment's influence on maternity care staff.

Two questionnaires applied the broadest approach in terms of construct completeness. The COSPOQ, which explored eleven dimensions, missing only *financial rewards*, and the MJSI which didn't include the dimension *physical work environment*. The internal consistency of the construct was not reported for the COSPOQ when used in study samples of midwives and the Cronbach's alpha was only acceptable (0.71) for the MJSI [31]. The LQWLQ-N also almost showed content completeness, missing only the categories *meaningfulness* and *health*.

In contrast to global satisfaction instruments or component scales, the strength of multidimensional instruments is to represent the whole construct of job satisfaction and determine the satisfaction in different domains. Thus, they identify correlations between the domains and may be an effective method for detecting changes in job satisfaction after interventions.

In addition to the completeness of the construct, the number of items is an important factor in the selection of a suitable instrument, and varies significantly in the instruments presented here. It should be critically noted that the larger the number of items, the greater the administrative effort and personal burden for users, so in-depth instruments may not be appropriate to measure job satisfaction on a regular basis. Most of the study instruments were translated from English into different languages and transferred from the Anglo-Saxon culture to other cultures without cultural adaptation, which may lead to decreased validity [74]. Consequently, the translation and validation process needs to apply not only linguistic adaptations, but instruments may well need to be adapted to the maternity care system in each particular country in a culturally appropriate manner.

Reliability characteristics were mentioned in most of the articles and we identified instruments with good reliability for use with midwives working in hospitals. Unfortunately, testretest reliability and sensitivity to change, which would be important to reflect the impact of interventions, were rarely, if at all, reported.

Future research should address all domains of midwives' job satisfaction to detect alternative opportunities for interventions to increase job satisfaction and midwives' intention to stay in the profession. It is hoped that this scoping review will aid future researchers in selecting an appropriate instrument.

## Strengths and limitations

The study approach included a comprehensive search strategy, and numerous assessment instruments in use for measuring job satisfaction of midwives were identified. The review was guided by the PRISMA-ScR extension. The instruments and their main characteristics are presented here, and the domains of importance for the assessment of midwives' job satisfaction identified. As this was a scoping review, the studies' methodological qualities were not critically assessed, which is considered a limitation [27]. The study instruments and the main quality criteria reported on here refer to a number of studies conducted in different countries with considerable differences in the maternity care system. This aspect must be taken into account

when assessing job satisfaction. Furthermore, some studies did not report the psychometric characteristics known to be relevant in the assessment of job satisfaction in midwives. Further research with instruments adapted to midwifery practice is required to enable methodological improvements in the study of job satisfaction of midwives.

## Conclusion

This review identified a number of questionnaires assessing midwives' job satisfaction. Only four instruments met the pre-set criteria for reliability and content validity for use in midwifery practice, so there is a need to develop or improve on instruments that capture all dimensions of midwives' job satisfaction in hospitals. Precise measurement tools are needed to evaluate interventions aimed at improving satisfaction. In view of the global shortage of midwives, it is vital that job satisfaction for midwives be improved in order to ensure both their retention in the workforce and high-quality midwifery care.

## Supporting information

**S1 Table. Preferred Reporting Items for Systematic reviews and Meta-Analyses extension for Scoping Reviews (PRISMA-ScR) checklist.**
(DOCX)

**S1 File. Search string.**
(DOCX)

## Acknowledgments

We like to thank Sue Travis for proofreading the article.

## Author Contributions

**Conceptualization:** Sonja Wangler, Gabriele Meyer, Gertrud M. Ayerle.

**Data curation:** Sonja Wangler, Joana Streffing.

**Investigation:** Sonja Wangler, Joana Streffing.

**Methodology:** Sonja Wangler, Gertrud M. Ayerle.

**Supervision:** Gabriele Meyer, Gertrud M. Ayerle.

**Visualization:** Sonja Wangler.

**Writing – original draft:** Sonja Wangler.

**Writing – review & editing:** Sonja Wangler, Anke Simon, Gabriele Meyer, Gertrud M. Ayerle.

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
