## [Decision Letter · Decision Letter 0]

17 Aug 2022

PONE-D-22-10981Measuring job satisfaction of midwives: A scoping reviewPLOS ONE

Dear Dr. Sonja,

Thank you for submitting your manuscript to PLOS ONE. After careful consideration, we feel that it has merit but does not fully meet PLOS ONE’s publication criteria as it currently stands. Therefore, we invite you to submit a revised version of the manuscript that addresses the points raised during the review process.

We look forward to receiving your revised manuscript.

Kind regards,

Negash Wakgari

Academic Editor

PLOS ONE

https://journals.plos.org/plosone/s/file?id=ba62/PLOSOne_formatting_sample_title_authors_affiliations.pdf".

Reviewers' comments:

Reviewer's Responses to Questions

**Comments to the Author**

1. Is the manuscript technically sound, and do the data support the conclusions?

Reviewer #1: Yes

Reviewer #2: Yes

2. Has the statistical analysis been performed appropriately and rigorously? 

Reviewer #1: Yes

Reviewer #2: I Don't Know

3. Have the authors made all data underlying the findings in their manuscript fully available?

Reviewer #1: Yes

Reviewer #2: Yes

4. Is the manuscript presented in an intelligible fashion and written in standard English?

Reviewer #1: Yes

Reviewer #2: Yes

5. Review Comments to the Author

Reviewer #1: I thought this was a well presented and clear scoping review. I wondered why the authors did not include the Journal name in Table 2? I do not have any suggestions to strengthen the paper further - the authors have presented a good scoping review.

Reviewer #2: Thank you for the opportunity to review this paper.

It is well written, unique, and provides sensible recommendations for expanding the development of tools that measure job satisfaction of midwives; a worthwhile endeavour.

My only suggestion is that more be written on the scoping review method under 'Methods'.

6. PLOS authors have the option to publish the peer review history of their article (what does this mean?). If published, this will include your full peer review and any attached files.

Reviewer #1: No

Reviewer #2: No

---

## [Author Response · Author response to Decision Letter 0]

13 Sep 2022

Dear editor, dear peer reviewers,

we appreciate the opportunity to revise our manuscript. Thank you very much for your peer reviews and appreciative feedback. 

Please find our point-by-point response below. We highlighted all changes in the manuscript. 

We hope that our manuscript now fulfils the expectations.

Reviewer 1

Comments to the author: I wondered why the authors did not include the Journal name in Table 2? 

 Thank you for this recommendation. We now present the journals’ names in Table 2.

Reviewer 2

Comments to the author: My only suggestion is that more be written on the scoping review method under 'Methods'.

 To clarify the method of the scoping review we added “We conducted a scoping review in order to explore the extent of the literature in the field of midwifery job satisfaction and to examine how research in this field is conducted. Scoping reviews aim to identify and map available evidence on an area of research in a transparent way. They bring together the evidence from heterogeneous sources and study approaches and can therefore detect research gaps in the existing literature.” (see lines 77-81.)

With kind regards,

Yours sincerely,

Sonja Wangler

on behalf of all co-authors

---

## [Editor Report · Decision Letter 1]

14 Sep 2022

Measuring job satisfaction of midwives: A scoping review

PONE-D-22-10981R1

Dear Ms. Sonja Wangler,

We’re pleased to inform you that your manuscript has been judged scientifically suitable for publication and will be formally accepted for publication once it meets all outstanding technical requirements.

Kind regards,

Negash Wakgari

Academic Editor

PLOS ONE
---

## [Editor Report · Acceptance letter]

19 Sep 2022

PONE-D-22-10981R1 

Measuring job satisfaction of midwives: A scoping review 

Dear Dr. Wangler:

I'm pleased to inform you that your manuscript has been deemed suitable for publication in PLOS ONE. Congratulations! Your manuscript is now with our production department. 

Kind regards, 

on behalf of

Mr. Negash Wakgari 

Academic Editor

PLOS ONE